

# Pressure-dependent performance of CEN-specified Condensation Particle Counters

Paulus S. Bauer[1], Dorian Spät[1], Martina Eisenhut[1], Andreas Gattringer[1], Bernadett Weinzierl[1]

[1]Aerosol Physics and Environmental Physics, Faculty of Physics, University of Vienna, 1090 Vienna, Austria

*Correspondence to*: Bernadett Weinzierl (bernadett.weinzierl@univie.ac.at)

**Abstract.** One of the most important parameters to quantify an aerosol is the particle number concentration. Condensation Particle Counters (CPCs) are commonly used to measure the aerosol number concentration in the nanometer range. To compare
the data from different measurement stations and campaigns it is important to harmonize the instrument specifications, which is why the Technical Specification CEN/TS 16976:2016 was introduced for CPCs. There, the parameters of the CEN-CPC are specified for standard pressure and temperature. However, CEN-CPCs are used in various surroundings, on high mountains or on airplanes, where they are exposed to low-pressure conditions. Here, we present the pressure-dependent performance (including the concentration linearity and counting efficiency) of two different models of CEN-CPCs, the Grimm 5410 CEN
and the TSI 3772-CEN. We found that their performance at 1000 hPa and 750 hPa was in accordance with the CEN-technical-specifications. Below 500 hPa, the performance decreased for both CPC-models, but the decrease was different for the two models. To gain insight into the performance of the two CPC-models, we performed a simulation study. This study included simulations of the saturation profiles and calculations of internal particle losses within the CPCs. The simulations reproduced the overall performance decrease with decreasing pressure and reveal that the internal structure of the CPC has a significant
influence on the performance. We anticipate our publication to provide a deeper understanding of the counting efficiency of CPCs and their pressure dependence. Our findings might be a starting point for new standards that include the pressure-dependent performance or they could help for designing new CPCs.



## 1 Introduction

Atmospheric aerosol substantially influences human health (Oberdörster et al., 2005; WHO, 2016) and our climate (IPCC, 2021, 2013). Therefore, it is constantly monitored either by ground-based measurement stations (Aerosol, Clouds and Trace Gases Research Infrastructure (ACTRIS), Global Atmosphere Watch (GAW) and many other ones, (Asmi et al., 2013; Rose et al., 2021)) or by aircraft measurements (IAGOS, the ATOM mission, A-LIFE and many other ones, (Bundke et al., 2015; Williamson et al., 2018; Brock et al., 2019; Kupc et al., 2018; Weinzierl et al., 2017). The particle number concentration is an
important parameter for quantifying the abundance of these short-lived atmospheric components. Condensation Particle Counters (CPCs) are commonly used to assess the number concentration directly in the nanometer range. To compare and evaluate the data from different monitoring stations and measurement campaigns it is important to harmonize the instrument specifications. Consequently, the Technical Specification CEN/TS 16976:2016 (published in 2016 and currently valid) was introduced for CPCs. This technical specification specifies the instrument specifications at standard pressure and temperature
(details in Sect. 1.2).

However, CPCs are operated under various environmental conditions including measurements at alpine monitoring stations, e.g. Sonnblick SBO (Rose et al., 2021) or the Himalayas (Bianchi et al., 2020), and in aircraft (Bundke et al., 2015; Williamson et al., 2018; Brock et al., 2019; Kupc et al., 2018). Due to the decreased ambient pressure at these altitudes the instruments' performance might change. Numerous studies have evaluated the pressure-dependent performance of CPCs (e.g. Takegawa
and Sakurai, 2011; Hermann et al., 2005; Hermann and Wiedensohler, 2001; Zhang and Liu, 1991; Bezantakos and Biskos, 2021; Mei et al., 2021), to the best of our knowledge no evaluation using CEN-specified CPCs has been conducted so far. Thus, the presented study aims at assessing the pressure-dependent performance of two models of CEN-CPCs, namely the Grimm 5410 CEN (in the following referred to as: Grimm CPC) and the TSI 3772-CEN (in the following referred to as: TSI CPC) (details on the CPCs in Sect. 1.2.1).

## 1.1 The counting efficiency of Condensation Particle Counters

Condensation particle counters represent one of the oldest measurement techniques in aerosol science and various different types of CPCs have been developed (McMurry, 2000). In this study we focused on continuous-flow thermal-diffusion type CPCs with an alcohol-based working fluid (McMurry, 2000). This means that the aerosol is continuously sampled by a CPC, consisting of a saturator, a condenser and a detection unit. For alcohol-based CPCs, the saturator is operated at a higher
temperature compared to the condenser, since the thermal diffusion rate (heat transfer) in air is higher than the diffusion rate of the (relatively large) alcohol molecules (Iida et al., 2009; McMurry, 2000; Hering and Stolzenburg, 2005). In the warm saturator the working fluid is vaporized and saturates the aerosol. Afterwards the aerosol enters the cold condenser where supersaturation is generated by thermal and vapor diffusion due to the rapid temperature change. Depending on the supersaturation, particle size and chemical composition, particles are activated by heterogeneous nucleation. By subsequent



condensational growth the activated particles grow to sizes large enough to be optically counted (e.g. with a laser) in the detector unit.

One important parameter to quantify the performance of a CPC is the counting efficiency $\eta_{CPC}$ as a function of the particle size $d_p$. According to Stolzenburg and McMurry (1991) the CPC counting efficiency can be decomposed into

$$\eta_{CPC}(d_p) = \eta_s(d_p) \cdot \eta_a(d_p) \cdot \eta_d(d_p), \quad (1)$$

where $\eta_s$ is the sampling efficiency, $\eta_a$ the activation efficiency and $\eta_d$ the detector efficiency inside the CPC. The sampling efficiency $\eta_s$ is determined by the sampling and transport losses inside the CPC. The activation efficiency $\eta_a$ accounts for the fraction of particles that are activated by heterogeneous nucleation in the condenser. Here it should be pointed out that the activation efficiency does not only depend on particle size, but also on the interactions between the particle and the vapor, e.g. solubility and wettability (Kupc et al., 2013). These chemical-dependent variables are not considered in this study, but are

discussed in other publications (e.g. Wlasits et al., 2020; Giechaskiel et al., 2011; Köhler, 1936). The detector efficiency $\eta_d$ comprises all activated particles that grow to droplets large enough to be measured by the optical detection system. In the original publication by Stolzenburg and McMurry (1991) the "detector efficiency $\eta_d$" is named "detection efficiency", however we renamed it to avoid confusions with the name detection efficiency, which is sometimes used synonymously for the counting efficiency $\eta_{CPC}$. In general, these three different efficiencies inside the CPC cannot be measured individually but can only be

evaluated theoretically using simulations.

The experimental approach to measure the counting efficiency $\eta'_{CPC}$ is to generate a monodisperse aerosol with a defined particle size $d_P$ (conventionally using a Differential Mobility Analyzer, DMA; multiple charged particles have to be taken into account). For the different particle sizes the detected concentration of the CPC, $N_{CPC}$, is then compared to the concentration of a reference instrument $N_{REF}$ (conventionally a Faraday Cup Electrometer, FCE):

$$\eta'_{CPC}(d_p) = \frac{N_{CPC}(d_p)}{N_{REF}(d_p)}. \quad (2)$$

The measured concentrations ($N_{CPC}$ and $N_{REF}$) contain also the transport efficiencies to the instruments and the size distribution of the aerosol (i.a. DMA transfer function), which must be considered for the exact counting efficiency (details in Stolzenburg and McMurry, 1991). Several measures were taken to minimize these effects described in the Methods section. As a result, the experimental counting efficiency is calculated as stated in Eq. (2) in this study.

The counting efficiency curve $\eta(d_p)$ has a very specific form with an increasing slope and a plateau region going from the small to the large particle sizes (see Fig. 3 for an example). It can be described by three important parameters, which are the plateau counting efficiency $\eta_\infty$, the cut-off diameter $d_{p,50}$ and the onset diameter $d_{p,0}$. Going from the largest sizes to the smallest, the plateau counting efficiency $\eta_\infty = \eta(d_p \rightarrow \infty)$ is the counting efficiency at large particle sizes (at least much larger than the cut-off diameter $d_{p,50}$, see next parameter) and is ideally $\eta_\infty = 100\%$ for the CEN-specified ambient conditions (see Sect. 1.2). The

cut-off diameter $d_{p,50}$ is the diameter where the counting efficiency reaches $50\%$, $\eta(d_{p,50}) = 50\%$. The cut-off diameter is conventionally seen as the lower detection limit of the measurement range of a CPC. The third parameter is the onset diameter $d_{p,0}$, which is the diameter of the smallest particles that are detected by the CPC, $\eta(d_{p,0}) \approx 0\%$. Since the onset diameter cannot





be obtained directly with these types of measurements, simulations or a fit function are utilized. The fit function including these three parameters is (Tuch et al., 2016; Stolzenburg and McMurry, 1991; ISO27891, 2015)

$$\eta(d_p) = \eta_\infty \cdot \left( 1 - \exp\left( -\frac{d_p - d_{p,0}}{d_{p,50fit} - d_{p,0}} \cdot ln2 \right) \right). \quad (3)$$


It should be pointed out that the fitted cut-off diameter $d_{p,50fit}$ ($\eta(d_{p,50fit})/\eta_\infty = 50\%$) is only equal to the previously defined cut-off diameter $d_{p,50}$ if the plateau counting efficiency is $\eta_\infty = 100\%$. The fitted cut-off diameter $d_{p,50fit}$ is shifted to lower diameters compared to the cut-off diameter $d_{p,50}$ if the plateau counting efficiency $\eta_\infty$ is lower than 100%.

To describe the increasing slope of the counting efficiency curve $\eta(d_p)$ we introduce the edge steepness parameter $\varepsilon$. Ideally

the steepness is the derivative of the counting efficiency curve $d\eta(d_p)/dd_p$ in the slope region. We approximate this derivative with the difference quotient between the onset and the cut-off diameter. We define the edge steepness parameter therefore as

$$\varepsilon = \frac{\Delta\eta(d_p)}{\Delta d_p} = \frac{50\% - 0\%}{d_{p,50} - d_{p,0}}, \quad (4)$$

where $d_{p,0}$ is the onset diameter derived from the fit and $d_{p,50}$ is the cut-off diameter. In principle, the edged steepness could be calculated with the fitted cut-off diameter $d_{p,50fit}$. However, $d_{p,50fit}$ depends on the plateau counting efficiency $\eta_\infty$, which is why

we will not use $d_{p,50fit}$ for the edge steepness in this publication. In general, the edge steepness $\varepsilon$ represents the percentual increase of the counting efficiency per nanometer of particle size between 0% and 50% and thus a steeper slope will give a larger edge steepness $\varepsilon$.

## 1.2 CEN Technical Specification 16976

The technical specification CEN/TS16976, entitled "Determination of the particle number concentration of atmospheric

aerosol", was published in 2016. It covers a variety of related topics, ranging from the sampling inlet system to the specifications of the counting devices, which in this case is a CPC. Some specifications are based on the ISO27891 ("Aerosol particle number concentration — Calibration of condensation particle counters") published 2015. The most relevant specifications for the presented study are summarized in the following paragraphs:

The CPC has to be a full-flow CPC, which means that there is no internal dilution of the aerosol flow inside the CPC. The

volumetric flow rate should only deviate by 5% from the nominal volumetric flow rate. The working fluid has to be n-butanol. Silver nanoparticles generated by the evaporation/condensation method (Scheibel and Porstendörfer, 1983), have to be used for the verification of the counting efficiency. The particle concentration should be between 3000 cm$^{-3}$ and 10 000 cm$^{-3}$. However, it is not clearly specified if that concentration is determined using the CPC or the reference instrument and which carrier gas should be used (air or $N_2$). The linearity of the concentration of the CPC in the plateau region has to be measured

at one fixed particle diameter between 30 nm and 50 nm by varying the concentration. The linearity (and hence the plateau counting efficiency $\eta_\infty$) has to be $1 \pm 5\%$. The cut-off diameter has to be determined by a fit, Eq. (3), and $d_{p,50fit}$ should be at 7 nm $\pm$ 0.7 nm. Since the plateau counting efficiency should be nearly one, the fitted $d_{p,50fit}$ is almost equal to the cut-off $d_{p,50}$.



However, for some of the following measurements at low pressure the plateau is lower than one and thus both cut-off diameter parameters ($d_{p,50}$ and $d_{p,50fit}$) have to be considered.

In addition to the cut-off diameter, the diameter corresponding to a counting efficiency of 90% $d_{p,90}$ should be below 14 nm. The measurements of the counting efficiency should be done at two different temperatures (15°C and 30°C) and at two different pressures, one higher than 900 hPa and one 200 hPa lower compared to an unknown reference. At this point, the technical specification is vague as it lacks a reference for the lower pressure measurement and does not contain specifications for a setup capable of creating these low-pressure conditions.

The standard temperature $T_0 = 296.15$ K, the standard pressure $p_0 = 1013.25$ hPa, the equations for the mean free path $\lambda_{air}(T,p)$, the dynamic viscosity $\mu(T)$ and the Cunningham correction factor $C_c(d_p, \lambda_{air})$ are specified in the CEN/TS16976 and can be found in Wiedensohler et al. (2012).

### 1.2.1   CEN-specified Condensation Particle Counters

We tested two different models of CEN-specified CPCs: the Grimm 5410 CEN and the TSI 3772-CEN. Here the characteristics
of both CPC-models are presented in alphabetical order. The parameters of the two CPCs are presented in Table 1. Both CPC-models are specified as full-flow CPCs with n-butanol as working fluid and they require an external vacuum pump.

The Grimm 5410 CEN has a nominal flow rate of 0.6 L min$^{-1}$ controlled by a temperature stabilized critical orifice. The saturator temperature is $T_{sat} = 36$°C and the condenser temperature is $T_{con} = 17$°C. The Grimm saturator has a displacer rod in the center, which must be considered as an annular tube in simulations and particle loss calculations. Single particle counting
is possible up to $10^5$ cm$^{-3}$ with internal coincidence correction for the Grimm CPC (Manual Grimm 5410, 2020).

The TSI 3772-CEN has a nominal flow rate of 1 L min$^{-1}$ controlled by a critical orifice. The flow is internally split up into eight pathways each with a flow rate of 0.125 L min$^{-1}$ for the condenser and saturator part (Kangasluoma et al., 2014). The saturator temperature is $T_{sat} = 39$°C and the condenser temperature is $T_{con} = 18$°C. Single particle counting is possible up to 5 · $10^4$ cm$^{-3}$ with life-time coincidence correction for the TSI CPC (Manual TSI 3772-CEN, 2016).

## 2   Methods

### 2.1 Experimental Setup

Figure 1 shows a schematic of the setup that we used to characterize the performance of the CEN CPCs under low pressure conditions. According to the CEN/TS 16976:2016, silver nanoparticles were generated via the evaporation/condensation method (Scheibel and Porstendörfer, 1983). Silver was heated between 940°C and 1050°C in a tube furnace and subsequently
cooled with a Liebig water cooler (15°C). A dilution flow was added to adjust the number concentration and size distribution of the nanoparticles. The furnace flow and the dilution flow were operated with laboratory pressurized air. Each flow was equipped with a needle valve, a silica-gel dryer, a HEPA-filter and a mass flow meter (TSI 4140) for precise flow control. The relative humidity of the air supply was kept below 10%.



The particles were selected corresponding to their electrical mobility with a classification system consisting of a soft X-ray
charger (TSI 3087) and a custom-made Vienna-type differential mobility analyzer (DMA, presented in Winkler et al. (2008b)
and Wlasits et al. (2020), referred to as nano-DMA). A positive voltage was applied to select negatively charged particles with
an equivalent mobility diameter ranging from 4 nm to 30 nm. The sheath air flow of 25 L min$^{-1}$ was generated with a closed
loop flow system, including HEPA-filters, a silica-gel dryer, a critical orifice and a pump. The flow through the classification
system (often referred as $Q_a$ aerosol flow or $Q_s$ sample flow) was determined by the flow of the sampling system (mostly the
flow through the limiting orifice). The resolution of the DMA defined by the flow ratio (Flagan, 1999) was therefore nearly
constant at 1:10 for all measurements. Diffusional broadening inside the DMA was not considered as the relevant mobility
diameters were above 5 nm (Wlasits et al., 2020). Multiple charged particles were considered as described in the Experimental
Procedure Sect. 2.2.

For low-pressure measurements, particles must either be size-selected (mobility-selected) in the low-pressure region (Hermann
and Wiedensohler, 2001) or the monodisperse particles must be transferred into the low pressure region via a valve (Zhang
and Liu, 1991) or an orifice (Takegawa and Sakurai, 2011). We tested the valve and the orifice system, both resulting in similar
counting efficiencies (not shown in this publication). However, the setup with the orifice similar to Takegawa and Sakurai
(2011) yielded higher particle concentrations and better control regarding concentration and pressure, hence we used this setup.
We did not see any charging artifacts in either system (see discussion in Hermann and Wiedensohler (2001) and Takegawa
and Sakurai (2011)). The green part in Fig. 1 indicates where the relevant pressure conditions are. For measurements at ambient
pressure the critical orifice was replaced by a stainless-steel pipe and the sampling flow was adjusted to $2.0 \pm 0.1$ L min$^{-1}$. At
our measurement location in Vienna the average ambient pressure was 996hPa $\pm$ 15hPa during our measurement period which
is why we refer all ambient pressure stages to 1000 hPa. At the 750 hPa pressure stage (after the orifice) we measured a flow
rate of $1.9 \pm 0.1$ L min$^{-1}$ (in front of the orifice). For the pressure stages at 500 hPa and below, a critical pressure ratio $p_{after}/p_{before}$
$< 0.528$ (pressure after and before the orifice) (Wiggert et al., 2016; Rathakrishnan, 2017) was sustained, ensuring a stable
flow of 2.1 L min$^{-1}$.

After the orifice, an aluminum mixing chamber assured that the aerosol was relaxed and well mixed before the splitter. The
"adjustment flow", which exits the mixing chamber halfway, controls the pressure stages. The stainless steel Y-splitter
guaranteed equal splitting of the aerosol flow for the detection instruments, which was verified by swapping the instruments
from one outlet to the other. From the DMA to the Y-splitter, every part and connection consisted of metal and the
instrumentations were attached with conductive tubing of equal length to reduce electrostatic deposition and ensure similar
transport losses for all instruments.

As a reference instrument we used a Faraday Cup Electrometer (FCE) from TAPCON (Winkler et al., 2008b, a). The
volumetric flow rate of the FCE was set to the nominal flow rate of the CPC ($Q_{Grimm}$ = 0.6 L min$^{-1}$, $Q_{TSI}$ = 1 L min$^{-1}$) using an
Alicat Mass Flow Controller (MC-Series). The pressure sensor (precision $\pm$0.1%) of the mass flow controller was also used
as reference instrument for our pressure stages (green part in Fig. 1). A pressure gauge [P] (Jumo Delos SI, precision
$\pm$0.35%) measured the pressure before the Agilent pump, which was max. 50 hPa. This assured that the flow of the CPC's



critical orifice stayed choked ($p_{after,CPC}/p_{before,CPC} < 0.528$, (Wiggert et al., 2016; Rathakrishnan, 2017)) for all pressure stages, down to 150 hPa. With the Jumo pressure gauge we also verified the pressure stability at various points between the limiting
orifice and the instrumentation, which was consistent with the pressure reading of the mass flow controller.

## 2.2 Experimental Procedure

Before we started with the counting efficiency measurements, we analyzed the size distribution (in the SI) of the silver nanoparticles by operating the DMA in scanning mode. We adjusted the temperatures and flows of the furnace to produce a minimum particle concentration of 2000 cm⁻³ and a maximum of 20 000 cm⁻³ measured by the FCE in the mobility size range
between 4 nm and 30 nm. CEN/TS 16976:2016 restricts the maximum particle concentration to 10 000 cm⁻³, however it is not clearly specified whether this maximum concentration applies to the CPC or the FCE. Furthermore, both CPCs are built to measure particle concentrations of up to 50 000 cm⁻³ (TSI 3772 CEN) or even beyond (Grimm 5410 CEN), when coincidence correction is switched on. Both CPC-models come with an internal coincidence correction (e.g. *live-time correction* for TSI 3772 CEN), which is why we chose the corrected concentration output of the CPCs ($N_{CPC}$) for our data analysis.
To avoid multiply charged particles, the size distribution was adjusted so that the mode of the distribution was lower than 30 nm and the concentration at 30 nm was on the lower end (slightly above 2000 cm⁻³). For the linear response and concentration comparison between CPC and FCE, the CEN/TS 16976:2016 recommends mobility particle sizes of $40 \pm 10$ nm. We used 30 nm particles selected from the right flank of the size distribution, where they can be consider as singly charged particles (Tuch et al., 2016; Wiedensohler, 1988).
The experimental procedure for the counting efficiency measurements was automated and always started with a two-minute zero measurement (0V at the DMA) to set the reference for the FCE and check the zero counts of the CPC. Then we alternately set zero volts for one minute and the voltage for the desired mobility diameters for two minutes at the DMA. The data (about 10 to 20 s) before and after each voltage transition was removed before taking the average, because of spikes in the concentration while the voltage was ramped up or down (Takegawa and Sakurai, 2011). For the FCE, we took the mean of the
zero measurements before and after the two-minute interval and subtracted the result from the mean of the two-minute interval to correct for the FCE background. The uncertainties resulting from this procedure were analyzed with Gaussian error propagation. To account for day-to-day variations, we started each measurement day with a counting efficiency measurement at ambient pressure (referred as 1000 hPa) and checked that it was consistent with previous measurements with instruments of the same type. In addition, we checked the flow rate of the sheath air flow (in and out of the DMA) and the flow before and
after the DMA each time the flow or the pressure level was changed.
For the low-pressure measurements, the butanol supply of the CPCs was removed and the auto-fill mode was switched off to avoid pressure leakage or flooding of the CPC. The stability of the counting efficiency without butanol supply was checked for both CPC-models by monitoring the efficiency of 30 nm particles over a long period of several hours similar to Takegawa and Sakurai (2011). Even for the lowest pressure settings where butanol diffusion and hence butanol losses are the largest, we
were able to measure more than 6 hours without any change in the counting efficiency. Despite these results, we filled up the



CPCs with butanol after each counting efficiency measurement routine under low-pressure to assure equal conditions for each measurement.

## 2.3  Simulation Methods

A simulation of the CPC is needed to investigate the individual efficiencies from Eq. (1) leading to the total counting efficiency
measured experimentally. There are many publications simulating the condenser or the whole CPC to investigate these efficiencies (e.g. Stolzenburg and McMurry, 1991; Zhang and Liu, 1990; Hering and Stolzenburg, 2005; Giechaskiel et al., 2011; Reinisch et al., 2019). The heart of these simulations are the equations for heat and mass transfer to calculate the temperature and vapor pressure profiles inside the CPC. The temperature profiles from a tube, where the wall temperature makes a sudden step change similar to in a CPC, is known in literature since the end of the 19[th] century as Graetz-Nusselt
problem (Eckert and Drake, 1972; Bird et al., 2002). The solutions to the Graetz-Nusselt problem is often used as reference for the simulations (e.g. Giechaskiel et al., 2011; Reinisch et al., 2019).

For our simulations we have made several assumptions: The problem is cylindrical symmetric so we consider only the axial ($z$) and radial ($r$) direction. We normalized the axial $z' = z/R_t$ and radial $r' = r/R_t$ distance with the radius of the tube $R_t$. The flow is incompressible, laminar, and has a fully developed parabolic flow profile $v(r') = 2\overline{v}(1 - r'^2)$, where $\overline{v} = Q/\pi R_t^2$ is
the average velocity of the flow and $Q$ the volumetric flow rate [m³ s⁻¹]. We do not consider any diffusion in axial direction nor coupling effects between mass and thermal diffusion nor effects from Stefan flow (Stolzenburg and McMurry, 1991). Heat and mass transfer onto the growing droplets are negligible (Zhang and Liu, 1990) and particle-particle interactions are neglected (Stolzenburg and McMurry, 1991). Thus, the results from the simulation are most accurate for monodisperse aerosol with a low particle concentration to neglect vapor depletion effects.
With these assumptions, the equations for heat and mass transfer can be reduced to

$$Pe_\psi \cdot \left(1 - r'^2\right)\frac{\partial \psi}{\partial z'} = \frac{1}{r'}\frac{\partial}{\partial r'}\left(r'\frac{\partial \psi}{\partial r'}\right) + \frac{\partial^2 \psi}{\partial z'^2} \ , \quad (5)$$

where $Pe$ is the Péclet number and $\psi$ is a placeholder variable that could either be the temperature $T$ or the partial vapor pressure $p_v$ for the heat and mass transfer equations, respectively. The derivation of Eq. (5) can be found in the supplementary material or partly in Bird et al. (2002). To solve this partial differential equation we used the FEniCS computer platform (Alnæs
et al., 2015).

The physical properties of the gas, the butanol vapor and the operating parameters of the CPC are incorporated in the Péclet number. The Péclet number $Pe$ is a dimensionless number comparing the advective and the diffusive transport rate. For thermal processes, the Péclet number $Pe_T$ can be decomposed into the dimensionless Reynolds $Re$ and Prandtl number $Pr$

$$Pe_T = Re \cdot Pr = \frac{\overline{v} \cdot 2R_t}{\nu} \cdot \frac{\nu}{\alpha} = \frac{\rho \cdot \overline{v} \cdot 2R_t}{\mu} \cdot \frac{\mu \cdot c_p}{k_T} \ , \quad (6)$$

where $\nu$ [m² s⁻¹] is the kinematic viscosity ($\nu = \mu/\rho$), $\mu$ [Pa s] the dynamic viscosity, $\rho$ the density of the gas [kg m⁻³], $\alpha$ [m² s⁻¹] the thermal diffusivity ($\alpha = k_T/(\rho \cdot c_p)$), $k_T$ [W m⁻¹ K⁻¹] the thermal conductivity and $c_p$ [J kg⁻¹ K⁻¹] the specific heat capacity





at constant pressure. For the partial vapor pressure, the Péclet number $Pe_{Pv}$ can be decomposed into the dimensionless Reynolds $Re$ and Schmidt number $Sc$

$$Pe_{p_v} = Re \cdot Sc = \frac{\overline{v} \cdot 2R_t}{\nu} \cdot \frac{\nu}{D_v} = \frac{\rho \cdot \overline{v} \cdot 2R_t}{\mu} \cdot \frac{\mu}{\rho \cdot D_v} \quad , \qquad (7)$$

where $D_v$ [m² s⁻¹] is the (binary) diffusion constant of the vapor in air. In Zhang and Liu (1990) it is shown that the Prandtl number $Pr$ and the Schmidt number $Sc$ depend only on temperature and not on pressure. However, the Reynolds number can be written as

$$Re = Re_0 \left(\frac{Q}{Q_0}\right)\left(\frac{p}{p_0}\right) \quad , \quad (8)$$

where $Re_0$, $Q_0$ and $p_0$ are the Reynolds number, the volumetric flow rate and the pressure at the standard operation conditions.

This implies that either a reduction of the pressure by some factor or a reduction of the volumetric flow rate by the same factor results in the same heat and mass transfer equations (Zhang and Liu, 1990). We will focus only on the pressure dependence in this publication.

To solve the partial differential Eq. (5) several boundary conditions are necessary. In most CPC simulation studies (e.g. Zhang and Liu, 1990; Hering and Stolzenburg, 2005; Giechaskiel et al., 2011), only the condenser of the CPC is simulated. However,

we included the insulator between the saturator and condenser in our simulations similar to Reinisch et al. (2019) which is of importance especially for the low-pressure cases. In the insulator, the wall temperature $T_{wall}$ is linearly decreasing from the saturator temperature $T_{sat}$ to the condenser temperature $T_{con}$. In the condenser the wall temperature $T_{wall}$ is constant at the condenser temperature $T_{con}$. The partial vapor at the wall $p_{wall} = p_{sat}(T_{wall})$ is set to the saturation vapor pressure at wall temperature. The incoming aerosol has the temperature of the saturator $T_{sat}$ and is considered to be fully saturated with butanol

vapor (Reinisch et al., 2019). This is especially true for the low-pressure case because the molecular diffusion is enhanced if the pressure is reduced and hence the aerosol gets saturated more easily. The vapor pressure of the incoming aerosol $p_v = p_{sat}(T_{sat})$ is set to the saturation vapor pressure at saturator temperature.

With the resulting temperature $T$ and partial vapor pressure $p_v$ profiles we calculated the saturation ratio profiles, $S = p_v/p_{sat}(T)$, where $p_{sat}(T)$ is the saturation vapor pressure at temperature $T$. Each point of the saturation ratio profile S and temperature

profile $T$ can be linked to an equilibrium diameter $D_{K,eq}$ via Kelvin theory

$$D_{K,eq} = \frac{4 \, \sigma_s \, M_w}{\rho_l \, R \, T \ln S} \quad , \qquad (9)$$

where $\sigma_s$, $M_w$ and $\rho_l$ are the surface tension, the molecular weight and the density of the condensing fluid (in our case liquid butanol), $R$ the universal gas constant and $T$ the absolute temperature (Winkler and Wagner, 2022). This so-called Kelvin diameter represents the minimal particle size that gets activated at the conditions present around the particle.

To calculate the activation efficiency $\eta_a$ from the simulations we discretized the profiles into axial $K_{ax}$ and radial $K_{rad}$ bins. Then the Kelvin diameter was calculated for each bin. For each particle size $d_p$ and for each radial bin (with the normalized radius $r_i'$) we determined if and where (in axial direction) the particle gets first activated. With this information, we calculated



the concentration of activated particles $N_{act}(d_p,r_i)$ for each particle size and radial bin, which we compared to the incoming number concentration $N_{in}(d_p,r_i)$. We then computed the activation efficiency $\eta_a$ with (Giechaskiel et al., 2011; Reinisch et al.,

280    2019):

$$\eta_a(d_p) = \frac{\sum_{i=1}^{K_{rad}} r_i'(1 - r_i'^2)\, N_{act}(d_p,r_i')}{\sum_{i=1}^{K_{rad}} r_i'(1 - r_i'^2)\, N_{in}(d_p,r_i')}. \quad (10)$$

Here the factor *(1-$r_i$'²)* accounts for the flow profile and the factor *$r_i$'* for the increase of the bin size and hence particle number concentration for each bin with the radial position *$r_i$'* (Reinisch et al., 2019).

To investigate the total counting efficiency of Eq. (1), we examined the sampling efficiency $\eta_s$, which includes the particle

losses from the inlet of the CPC to its condenser. For the size range below 100 nm we considered only diffusional losses and therefore we implemented Eq. (21) and Eq. (22) from Weiden et al. (2009) for cylindrical tubes. For the particle losses in the annular saturator of the Grimm CPC we used the formula from Talebizadehsardari et al. (2020). The detector efficiency $\eta_d$ includes the growth of the particles to optical sizes, which we have analyzed with a growth model (in the supplementary material). To conclude, all activated particles can be considered optically detectable due to the rapid growth (details in (Hering

and Stolzenburg, 2005; Giechaskiel et al., 2011). Neither losses in the focusing region nor in the optic section of the CPC were studied in this publication and thus we set the detector efficiency $\eta_d = 1$.

## 3    Results and Discussion

### 3.1  Experimental Results

We analyzed four identically-constructed Grimm 5410 CEN CPCs and two identically-constructed TSI 3772-CEN CPCs. We

conducted measurements at the pressure levels of 1000 hPa, 750 hPa, 500 hPa, 375 hPa (only TSI CPCs), 250 hPa, and 150 hPa (only Grimm CPCs). In the first set of experiments, we analyzed the linearity of the CPCs compared to the reference FCE, shown in Fig. 2. In a second set of experiments, the size-dependent counting efficiency curves for different pressure stages are shown in Fig. 3. The results of the fits performed in Fig. 2 and Fig. 3 are reported in Table 2. For clarity, Fig. 2, Fig. 3, and Table 2 are presented without error bars or uncertainties, but detailed figures and tables with the necessary information are

available in the supplementary information.

The linear response of the CPC is very important for the later counting efficiency measurements. We analyzed the linearity at 30 nm for one CPC of each CPC-model. In Fig. 2 the concentration of the CPCs (with coincidence correction switched on) is compared to the reference concentration of the FCE for the different pressure stages in a log-log plot. The results of the linear fit through the origin are reported in the column *CPC/FCE* in Table 2. For clarity the 750 hPa points and fits are not shown in

Fig. 2, but are plotted in the supplementary material.

At 1000 hPa and 750 hPa, the linearity of both CPC-models is in agreement with the CEN-standard (1 ± 5%, see Sect. 1.2). At 500 hPa and 250 hPa the response of both CPC-models was still linear but below 1. This linear behavior (below $2 \cdot 10^4$ cm$^{-3}$)





prompted us to utilize the coincidence corrected concentration of the CPCs for the counting efficiency measurements. Interestingly, at 500 hPa the linear fit results for both CPC-models are around 93 %, whereas at 250 hPa the Grimm CPC is still above 80% and the TSI CPC is around 55%. Therefore, we have added the 375 hPa pressure stage for the TSI CPC and the 150 hPa for the Grimm CPC to investigate this drop in plateau counting efficiency. Since the Grimm CPC showed some non-linear behavior at 150 hPa for particle concentrations above 6 000 cm$^{-3}$, we only considered FCE concentrations below 6 000 cm$^{-3}$ for the linear fit.

The size-dependent counting efficiency measurements were conducted for four Grimm CPCs and two TSI CPCs at the different pressure stages, except for 150hPa which was only measured for three different Grimm CPCs. The resulting counting efficiencies curves were very consistent between the different CPCs of each CPC-model (see supplementary material). Thus, we averaged the counting efficiency measurements for each CPC-model, which is shown in Fig. 3. The counting efficiency curves were fitted with Eq. (3). The resulting parameters $\eta_\infty$, $d_{p,0}$ and $d_{p,50fit}$ are reported in Table 2 along with the subsequently obtained parameters $d_{p,50}$ and $d_{p,90}$ (see Sect. 1.1). The edge steepness $\varepsilon$ calculated with Eq. (4) was added to quantify the slope of the counting efficiency curves. A higher value of the edge steepness $\varepsilon$ represents a steeper slope.

The results from the linearity fit of Fig. 2 (CPC/FCE in Table 2) and the results of $\eta_\infty$ from the counting efficiency fit of Fig. 3 ($\eta_\infty$ in Table 2) represent a measure of the plateau counting efficiency. Both representations of the plateau counting efficiency agree to each other within 5% for pressures down to 250 hPa for the Grimm CPCs and down to 375 hPa for the TSI CPCs. This is notable since the plateau counting efficiency is determined in two different ways, with different fitting functions and procedures. In addition, the linearity of Fig. 2 is only measured with one CPC whereas the data of Fig. 3 is the average of an ensemble of CPCs. Below the mentioned pressure stages, the plateau counting efficiencies (*CPC/FCE*, $\eta_\infty$) of both CPC-models show a bigger difference, which originates from various effects including the nonlinear behavior of the CPCs, the averaging, and the flat shape (large edge steepness ε) of the counting efficiency fit in Fig. 3, which shifts $\eta_\infty$ to higher values. In general, there is a big change in the counting efficiency curves and fit-parameters (see Fig. 3 and Table 2) from the respective second lowest pressure stage (250 hPa for the Grimm CPCs and 375 hPa for the TSI CPCs) to the lowest pressure stage (150 hPa for the Grimm CPCs and 250 hPa for the TSI CPCs) for both CPC-models.

Both CPC-models show a similar trend regarding the pressure dependence of the counting efficiency, which is comparable to Takegawa and Sakurai (2011) and Zhang and Liu (1990). For decreasing pressure, the plateau counting efficiency $\eta_\infty$ is decreasing, the cut-off diameter ($d_{p,50fit}$ and $d_{p,50}$) is increasing and the edge steepness $\varepsilon$ is decreasing (the curves are getting flatter). For 1000 hPa and 750 hPa, the Grimm CPCs and the TSI CPCs are in agreement with the CEN-technical-specifications (see Sect. 1.2). At 500 hPa and below, the Grimm CPCs generally have a higher plateau counting efficiency and a lower cut-off diameter than the TSI CPCs. A further difference between the CPC-models is the pressure-dependence of the onset diameter $d_{p,0}$. For the Grimm CPCs the onset diameter changes only marginally with pressure, which is why the curves in Fig. 3 seem to emerge from one point. For the TSI CPCs the onset diameter is increasing with decreasing pressure, which is why the curves look more separated. To investigate this difference in behavior between both CEN-CPC-models we conducted simulations, which are presented in the next section.



## 3.2 Numerical Results

We simulated the temperature and the vapor pressure profile for the insulator and condenser of both CPC-models as described in Sect. 2.3. From the results we calculated the saturation ratio S depicted as contour plots in Fig. 4 for 1000 hPa and 250 hPa.

The corresponding centerline profiles (at $r = 0$) of the saturation ratio S, the saturation vapor pressure and the partial vapor pressure for 1000 hPa and 250 hPa are displayed in Fig. 5. The simulations were performed for all pressure stages of the experimental results (Sect. 3.1), but for clarity only the profiles for 1000 hPa and 250 hPa are provided. In all plots the x-axis represents the normalized length $z' = z/R_t$ (see Sect. 2.3). To visualize the different lengths of the CPC-model's insulator and condenser the x-axes are set to the same scale. The length of the insulator is indicated either with a vertical dashed line (Fig.

4) or with a gray shaded part (Fig. 5). The black 7-nm-line in Fig. 4 encloses the area where the supersaturation is sufficient to activate at least 7 nm particles (Eq. (9)). The black 7-nm-line corresponds to the desired cut-off diameter $d_{p,50} = 7$ nm of the CEN standard (see Sect. 1.2).

From the simulated saturation ratio profiles, we calculated the activation efficiency $\eta_a$ (Eq. (10) in Sect. 2.3) for the different pressure stages shown in Fig. 6. Combined with the sampling efficiency $\eta_s$, which includes the particle losses of the inlet and

saturator of the CPC, we obtain the numerically calculated counting efficiencies presented in Fig. 7. In both figures the measured counting efficiencies of Fig. 3 were added as reference. Table 3 presents the parameters evaluated from the numerically calculated counting efficiency (Fig. 7), the counting efficiency at $d_p = 30$ nm ($\eta(30$ nm$)$) and the other parameters specified in Sect. 1.1 (onset $d_{p,0}$, cut-off $d_{p,50}$, 90%-diameter $d_{p,90}$ and edge steepness $\varepsilon$).

The profiles (Fig. 4 and Fig. 5) and the corresponding parameters give insights in the behavior of the different CPC-models.

One of the most distinct differences between the two CPC-models are the relative lengths of the insulator and condenser. The Grimm CPC has a shorter insulator and a longer condenser than the TSI CPC relative to the tube radius. But, the TSI CPC has a lower flowrate in this section of the instrument (see Table 1), which has to be considered for the saturation profile S (see Eq. (8) in Sect. 2.3). First, we compare the saturation profiles of both CPC-models at 1000 hPa. One important characteristic is the point with the highest (super) saturation ratio $S_{max}$, which is relevant for the smallest particles that get activated (see Kelvin-

Eq. (9)) classified by the onset diameter $d_{p,0}$. For the Grimm CPC $S_{max}$ is located almost at the end of the condenser, whereas for the TSI CPC it is close to entrance of the condenser. For the TSI CPC the saturation profile (especially the black 7-nm-line) and vapor pressure profile reach significantly into the insulator. This demonstrates the importance to include the insulator into simulations as stated in Reinisch et al. (2019).

At 250 hPa the saturation profile and $S_{max}$ of both CPC-models is shifted to the left (towards the entrance). This is in good

agreement with theoretical considerations of Eq. (8) in Sect. 2.3. For the Grimm CPC $S_{max}$ is almost as high as at 1000 hPa, which results only in a small shift of the onset diameter (see Fig. 6 and Fig. 7). For the TSI CPC at 250 hPa the saturation profile is substantially different compared to the 1000 hPa. It is moved into the insulator part and $S_{max}$ is even lower than required for activating 7 nm particles (no black 7-nm-line). Thus, the onset diameter of the TSI CPC is beyond 7 nm at 250 hPa. The onset diameter shift also explains why the activation efficiency curves $\eta_a$ in Fig. 6 fall nearly onto each other for the



Grimm CPC and for the TSI CPC the curves and onset diameters are shifted for each pressure stage. Interestingly, the onset diameters $d_{p,0}$ of both CPC-models are in general lower for the measured counting efficiency curves (see Table 2) compared to the numerically calculated ones (Table 3). The difference between the calculated and measured onset diameter might be explained by the chemical interplay of butanol nucleating onto silver particles, which is responsible for the activation of particles smaller than the Kelvin-diameter (Eq. (9)) (Tauber et al., 2019). This chemical effect is depending on the nucleation

temperature and is increasing with decreasing nucleation temperature. Further investigations and a parametrization are needed to incorporate this chemical effect into simulations.

In Fig. 6, for all pressure stages the calculated activation efficiency $\eta_a$ reaches 100% at some particle diameter. The required supersaturation to activate a particle is decreasing exponentially with increasing Kelvin diameter Eq. (9). In the simulations, for particles with a certain diameter (and larger) this supersaturation is reached for each path in the saturator. Thus, the

calculated activation efficiency $\eta_a$ solely cannot reflect the decreasing plateau counting efficiency $\eta_\infty$ with decreasing pressure. Additionally, the edge steepness $\varepsilon$ of the calculated activation efficiency $\eta_a$ stays almost the same for all pressure stages. Combined with the sampling efficiency $\eta_s$ (Fig. 7), which takes the particle losses inside the inlet and saturator of the CPC into account, some of the decreasing plateau counting efficiency as well as some of the decreasing edge steepness $\varepsilon$ (curves get flatter) can be explained. For the Grimm CPC, also the onset diameter is slightly shifted by the particle losses. Comparing the

parameters of the simulations (Table 3) with those of the measurements (Table 2), the onset diameters $d_{p,0}$ and the cut-off $d_{p,50}$ for the TSI CPC are mostly within 0.5 nm. For the Grimm CPC, there is a larger deviation, which might be explained by the chemical effect on the onset diameter, because in the Grimm CPCs the nucleation temperature is slightly lower and therefore the effect is more relevant (Tauber et al., 2019). For both CPCs, at 500 hPa the parameters and counting efficiency curves of the simulations fit best with those of the measurements. In general, the calculated counting efficiency ($\eta_{calc} = \eta_a \cdot \eta_s$) captures

the pressure dependent behavior of both CPC models quite well except for the strong decline of the plateau counting efficiency $\eta_\infty$ at the lowest pressure stage.

For the pressure-dependent shift of the plateau counting efficiency $\eta_\infty$ other factors should also be considered. The losses inside the reference instrument, the FCE, influences the results of the measurements, but they are not included in the simulations (this can best be seen by the difference of $\eta_\infty$ (Table 2) and $\eta(30$ nm$)$ (Table 3) at 1000 hPa). Another important factor for $\eta_\infty$ might

be the losses during growth of activated particles and the losses inside the optics, which are summarized in the detector efficiency $\eta_d$. This detector efficiency $\eta_d$ also includes the focusing of the particles into the optics, which was partly investigated by Takegawa and Sakurai (2011). Further research with measurements and (more advanced) simulations are needed to evaluate the pressure dependence of the detector efficiency $\eta_d$ which is beyond the scope of this publication.



## 4 Conclusion

In this study, we evaluated the pressure-dependent performance of two CEN/TS 16976:2016 standardized CPC-models, the Grimm 5410 CEN and the TSI 3772-CEN. The performance of four Grimm CPCs and two TSI CPCs was analyzed at 1000 hPa, 750 hPa, 500 hPa and 250 hPa. Additionally, we added measurements at 375 hPa for the TSI CPCs and at 150 hPa for the Grimm CPCs. In general, we found a similar trend for the pressure-dependent performance as shown in other publications (e.g. Takegawa and Sakurai, 2011; Zhang and Liu, 1991; Bezantakos and Biskos, 2021; Mei et al., 2021): With decreasing pressure the plateau counting efficiency $\eta_\infty$ is decreasing, the cut-off diameter $d_{p,50}$ is increasing and the edge steepness $\varepsilon$ is decreasing. At 1000 hPa and 750 hPa both CPC-models fulfill the CEN/TS 16976:2016 criteria. Below 500 hPa, the pressure-dependent performance differs between the two CPC-models; the Grimm CPCs have a higher plateau counting efficiency $\eta_\infty$ and a lower cut-off diameter $d_{p,50}$ than the TSI CPCs. The onset diameter $d_{p,0}$ stays almost constant for the Grimm CPCs.

To gain more insights in the different performance of the two CPC models, we conducted a simulation study. We simulated the temperature and vapor pressure profile inside the insulator and condenser of both CPC models. From this we calculated the activation efficiency $\eta_a$ and combined it with the sampling efficiency $\eta_s$, which includes the diffusional losses inside the CPC. The simulation results capture the overall pressure dependence of the experimental results. The simulations reveal that for the TSI CPCs the onset diameter $d_{p,0}$ and hence the counting efficiency curves shifts due to the reduction of the saturation ratio, whereas for the Grimm CPCs it mainly shifts because of diffusional losses. In addition, the decrease of the edge steepness $\varepsilon$ with decreasing pressure can only be explained by including the sampling efficiency $\eta_s$. We have not included the detector efficiency $\eta_d$ in our numerical study due to its complexity. To fully understand the pressure dependence of the counting efficiency, it is necessary to investigate the detector efficiency $\eta_d$ further which is beyond the scope of this study.

There are several approaches to eliminate the pressure-dependence of CPCs by changing the design (e.g. Williamson et al., 2018; Wilson et al., 1983) or the working-fluid (e.g. Hermann et al., 2005; Williamson et al., 2018) of the CPC or by altering the saturator and condenser temperatures (e.g. Hermann and Wiedensohler, 2001; Bezantakos and Biskos, 2021). Another approach to compensate for the pressure-dependent effects might be to utilize the implications of Eq. (8). If the volumetric flow inside the CPCs is increased by the same factor as the pressure is reduced, the temperature and vapor pressure profile inside the insulator and condenser should stay the same. This would result in an activation efficiency $\eta_a$ independent of the pressure (Zhang and Liu, 1991). However, the focusing into the optics and the optical counting is affected by the change in the volumetric flowrate. Therefore, the varying flowrate approach is not straight forward, but it might be interesting for further investigations.

For harmonizing the data from high-alpine measurement stations (ACTRIS, etc.) or aircraft measurements it might be of interest to include specifications on the pressure-dependent performance of CPCs in a new standard. Here our approach to separate the different efficiencies and the results of our measurements could help to correct for the pressure-dependent effects. Our findings might also be helpful for designing new CPCs.



**Author contribution**

PSB and BW designed and supervised the study. PSB, DS, and ME prepared the experiment, tested the experimental set-up
and performed the measurements. The experimental results are partly presented in the master thesis of ME. AG developed the
computer code and supporting algorithms for the instrumentation. PSB and DS analyzed the data with support from ME. DS
performed the simulations with supervision and contributions from PSB. The interpretation of the results is the outcome of
numerous discussions among PSB, DS and all co-authors. PSB wrote the manuscript with revisions from DS and BW. All co-
authors read and commented on the manuscript.

**Competing interests**

The authors declare that they have no conflict of interest.

**Acknowledgements**

The authors acknowledge the great support during our measurement campaign from Gerhard Steiner (Grimm), Sebastian
Schmitt (TSI) and Torsten Tritscher (TSI). The authors thank Tristan Reinisch (AVL DiTEST) for the discussion on the
simulations of CPCs. Many thanks to Peter Wlasits and Paul Winkler (both University of Vienna) for the great discussions
and their support and help with the instrumentation.

**Financial support.**

This project has received funding from the European Research Council (ERC) under the European Union's Horizon 2020
research and innovation framework program under grant agreement No. 640458 (A-LIFE), and the ESA project A-CARE
(ESA Contract No. 4000125810/18/NL/CT/gp).
The CPCs were purchased under the University of Vienna's investment program (IP734013), and through further support by
the University of Vienna.
Open access funding provided by University of Vienna.



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



**Table 1: Flowrates and Temperatures of the Grimm 5410 CEN CPC and the TSI 3772-CEN CPC. The butanol saturation vapor pressures for saturator and condenser are important for the simulations. They are calculated for the corresponding temperatures.**

| CPC-model | Nominal Inlet Flowrate | Internal Flowrate | Saturator Temperature | Condenser Temperature | Saturator $p_{sat}$ | Condenser $p_{sat}$ |
|---|---|---|---|---|---|---|
| Grimm 5410 CEN | 0.6 L min⁻¹ | 0.6 L min⁻¹ | 36 °C | 17 °C | 19.5 hPa | 5.1 hPa |
| TSI 3772-CEN | 1.0 L min⁻¹ | 8 x 0.125 L min⁻¹ | 39 °C | 18 °C | 23.7 hPa | 5.5 hPa |





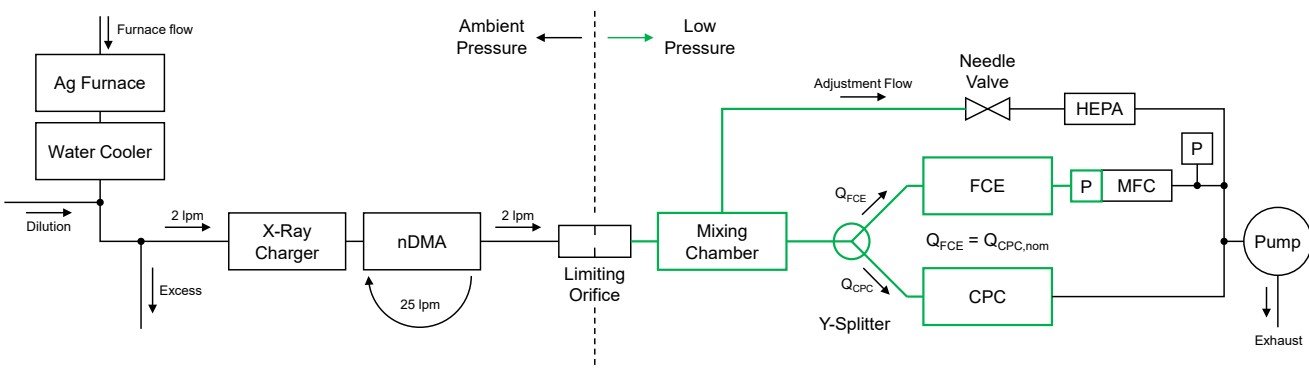


**Figure 1: Schematic setup to measure the counting efficiencies of CEN CPCs under low pressure conditions. The green part indicates where the relevant pressure conditions are. The limiting orifice was replaced by a stainless-steel tube during ambient pressure measurements. If the pressure ratio $p_{after}/p_{before}$ at the limiting orifice was smaller than 0.528, the limiting orifice acted as critical orifice with a constant flow rate of 2.1 l/min. The flow rate of the FCE ($Q_{FCE}$) was controlled by a mass flow controller (MFC) which**

**was set to the same volumetric flow rate as the nominal flow rate of the investigated CPC ($Q_{CPC,nom}$). The MFC pressure sensor (P at MFC) was used as reference for the different pressure settings (in the green part). A separate pressure gauge (P) measured the pressure in front of the pump.**




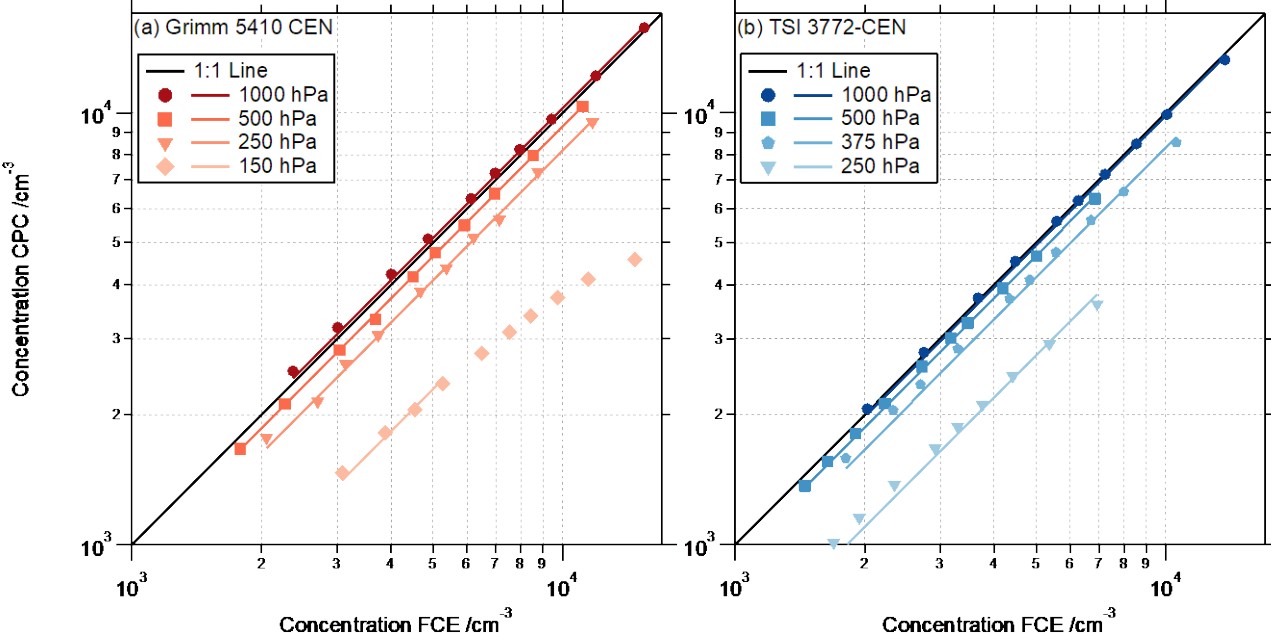

**Figure 2: Linearity analysis of the CPC concentration ((a) Grimm 5410 CEN and (b) TSI 3772-CEN) compared to the FCE reference concentration for 30 nm particles. The black solid line represents the ideal 1:1 line. The symbols represent the measurements at the different pressure stages and the corresponding lines are linear fits through the origin (fit results in column CPC/FCE in Table 2). The 750 hPa results lie very close to the 1000 hPa results for both CPC-models and are not shown here for clarity. The Grimm CPC (a) showed some non-linear response at 150 hPa and only data with FCE concentrations below 6000 cm$^{-3}$ were considered for the linear fit.**



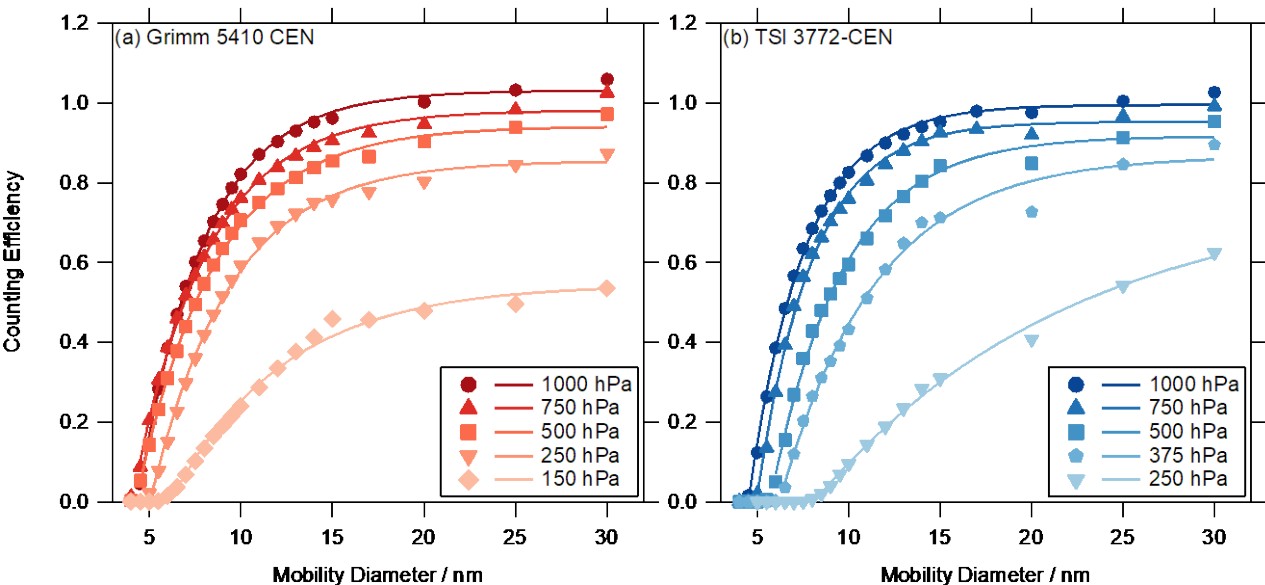

**Figure 3: Counting efficiency curves ((a) Grimm 5410 CEN and (b) TSI 3772-CEN) as a function of the mobility diameter. The counting efficiency was measured with four Grimm CPCs and two TSI CPCs. The markers represent the average over the ensemble of each CPC-model. The fit is defined by the Eq. (3) and the parameters are reported in Table 2.**






**Table 2: Results of the fits from Fig. 2 and Fig. 3 for the different CPC-models and pressure stages. The column CPC/FCE is the result of a linear fit through the origin of Fig. 2 comparing the concentration of the CPC to the FCE. The columns $\eta_\infty$, $d_{p,0}$ and $d_{p,50fit}$ represent the fitting parameters of the counting efficiency Eq. (3) of Fig. 3. The parameters $d_{p,50}$ and $d_{p,90}$ were calculated from the fitted counting efficiency parameters. The edge steepness $\varepsilon$ was calculated with Eq. (4). A table with the uncertainties of the fits is presented in the supplementary material.**

| CPC-model | Pressure [hPa] | CPC/FCE | $\eta_\infty$ | $d_{p,0}$ [nm] | $d_{p,50fit}$ [nm] | $d_{p,50}$ [nm] | $d_{p,90}$ [nm] | $\varepsilon$ [% nm$^{-1}$] |
|---|---|---|---|---|---|---|---|---|
| Grimm 5410 CEN | 1000 | 1.026 | 1.031 | 4.4 | 6.9 | 6.7 | 11.8 | 21.7 |
| Grimm 5410 CEN | 750 | 0.983 | 0.989 | 4.0 | 6.9 | 6.9 | 13.9 | 17.2 |
| Grimm 5410 CEN | 500 | 0.929 | 0.942 | 4.3 | 7.3 | 7.5 | 17.7 | 15.6 |
| Grimm 5410 CEN | 250 | 0.816 | 0.855 | 5.0 | 8.1 | 8.9 | - | 12.8 |
| Grimm 5410 CEN | 150 | 0.457 | 0.546 | 6.1 | 10.4 | 21.5 | - | 3.2 |
| TSI 3772-CEN | 1000 | 0.986 | 0.995 | 4.5 | 6.6 | 6.6 | 11.7 | 23.8 |
| TSI 3772-CEN | 750 | 0.973 | 0.953 | 4.9 | 7.0 | 7.2 | 13.7 | 21.7 |
| TSI 3772-CEN | 500 | 0.934 | 0.917 | 5.7 | 8.4 | 8.8 | 21.6 | 16.1 |
| TSI 3772-CEN | 375 | 0.831 | 0.867 | 6.2 | 9.8 | 10.7 | - | 11.1 |
| TSI 3772-CEN | 250 | 0.549 | 0.781 | 8.1 | 17.9 | 22.6 | - | 3.4 |





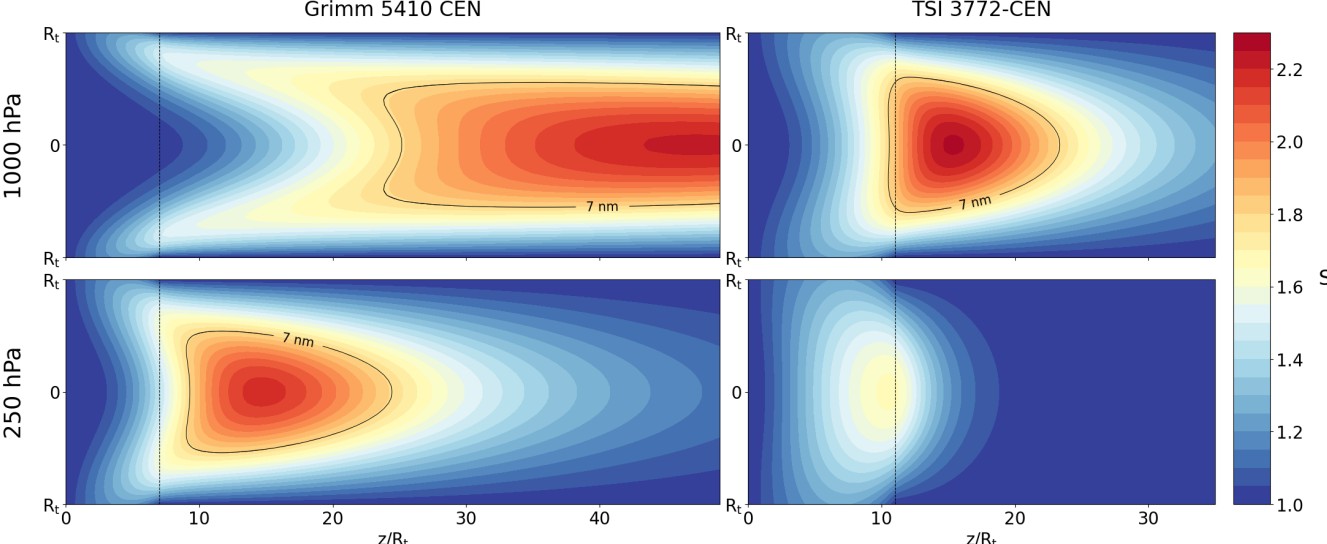


**Figure 4: Simulated saturation ratio S of the CPC's insulator and condenser. The left column shows the results for the Grimm 5410 CEN CPC and the right column for the TSI 3772-CEN CPC. The upper row represents the 1000 hPa case, the lower row the 250 hPa one. The x-axis represents the normalized length z' = z/$R_t$ and is set to the same scale for both CPC-models. The length of the insulator is marked with a vertical dashed line. The black 7-nm-line encloses the area where the supersaturation is sufficient to**

**activate particles with a Kelvin diameter of at least 7 nm (Eq. (9)). The centerline profiles (at r = 0) of the saturation ratio and the corresponding vapor pressures are presented in Fig. 5.**





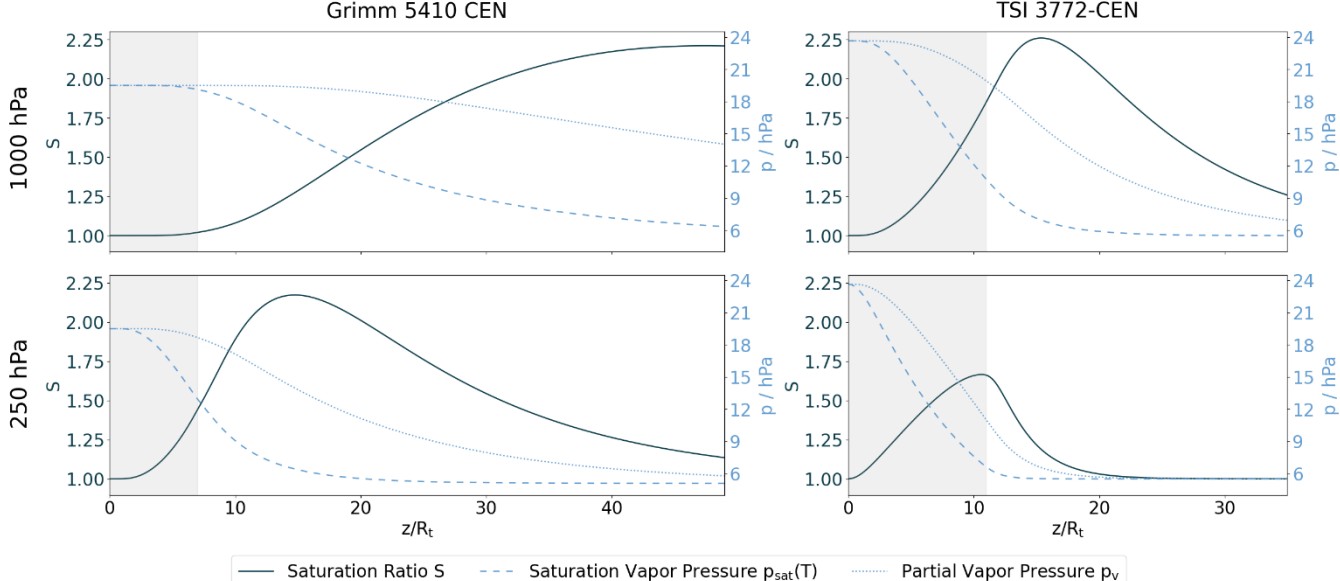

**Figure 5: Centerline profiles of the saturation ratio S, the saturation vapor pressure and the partial vapor pressure of butanol resulting from the CPC's insulator and condenser simulations (see Fig. 4). The left column shows the results for the Grimm 5410 CEN CPC and the right column for the TSI 3772-CEN CPC. The upper row represents the 1000 hPa case, the lower row the 250 hPa one. The x-axis represents the normalized length z' = z/R$_t$ and is set to the same scale for both CPC-models. The insulator part is shaded in grey. The y-axis of the saturation ratio S is on the left side, for the vapor pressures on the right one.**



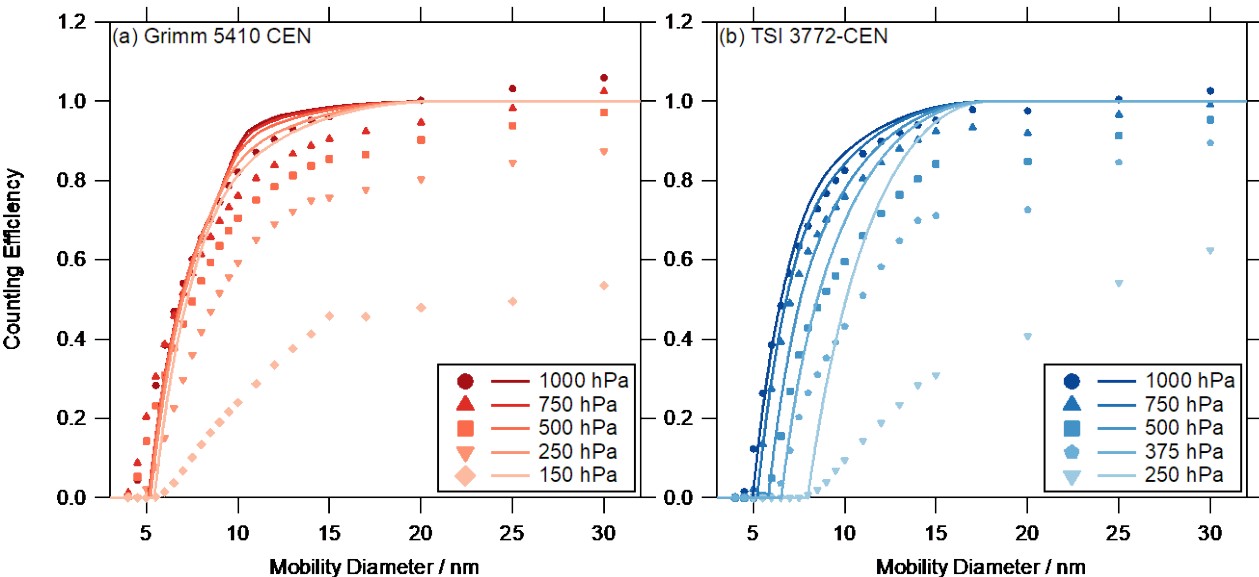

**Figure 6. Activation efficiency η_a curves (lines) ((a) Grimm 5410 CEN and (b) TSI 3772-CEN) calculated from the simulations (Fig. 4) for the different pressure stages. The markers are for reference and represent the results of the measured counting efficiencies (Fig. 3).**





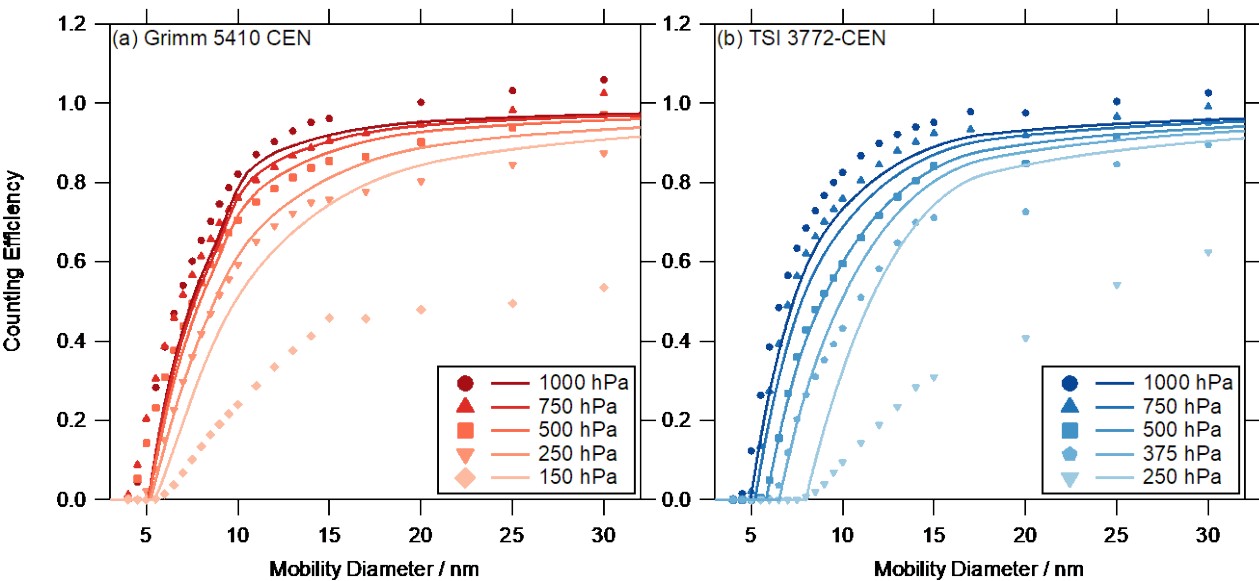

**Figure 7. Numerically calculated counting efficiencies (lines) ((a) Grimm 5410 CEN and (b) TSI 3772-CEN) including the activation efficiency $\eta_a$ (Fig. 6) and the sampling efficiency $\eta_s$ for the different pressure stages. The markers are for reference and represent the results of the measured counting efficiencies (Fig. 3). The plateau counting efficiency and the other parameters are listed in Table 3.**





**Table 3. Parameters of the numerically calculated counting efficiencies of Fig. 7 for the different CPC-models and pressure stages. The counting efficiency computed at $d_p$ = 30 nm is presented in the column η(30 nm). The various diameters ($d_{p,0}$, $d_{p,50}$ and $d_{p,90}$) were evaluated at the indicated counting efficiency values. The edge steepness ε was calculated with Eq. (4).**

| CPC-model | Pressure [hPa] | η(30 nm) | $d_{p,0}$ [nm] | $d_{p,50}$ [nm] | $d_{p,90}$ [nm] | ε [% nm⁻¹] |
|---|---|---|---|---|---|---|
| Grimm 5410 CEN | 1000 | 0.972 | 5.1 | 7.5 | 13.5 | 20.8 |
| Grimm 5410 CEN | 750 | 0.967 | 5.1 | 7.6 | 14.7 | 20.0 |
| Grimm 5410 CEN | 500 | 0.958 | 5.1 | 7.9 | 16.7 | 17.9 |
| Grimm 5410 CEN | 250 | 0.934 | 5.2 | 8.7 | 21.8 | 14.3 |
| Grimm 5410 CEN | 150 | 0.908 | 5.5 | 9.9 | 28.1 | 11.4 |
| TSI 3772-CEN | 1000 | 0.959 | 5.0 | 7.3 | 15.6 | 21.7 |
| TSI 3772-CEN | 750 | 0.951 | 5.2 | 7.8 | 17.0 | 19.2 |
| TSI 3772-CEN | 500 | 0.937 | 5.8 | 8.9 | 20.6 | 16.1 |
| TSI 3772-CEN | 375 | 0.925 | 6.5 | 9.9 | 23.7 | 14.7 |
| TSI 3772-CEN | 250 | 0.904 | 8.0 | 11.5 | 29.0 | 14.3 |

660





## Appendix A: List of symbols

Most important symbols in order of occurrence

| Symbol | Parameter |
|---|---|
| $\eta_{CPC}$ | CPC counting efficiency |
| $\eta_s$, $\eta_a$, $\eta_d$ | Sampling, activation and detector efficiency |
| $\eta'_{CPC}$ | Measured counting efficiency |
| $\eta_\infty$ | Plateau counting efficiency |
| $d_{p,50}$ | Cut-off diameter |
| $d_{p,0}$ | Onset diameter |
| $d_{p,50fit}$ | Fitted cut-off diameter |
| $\varepsilon$ | Edge steepness parameter $\varepsilon = \Delta\eta(d_p)/\Delta d_p$ |
| $T$, $T_{sat}$, $T_{con}$, $T_{wall}$ | Temperatures, saturator, condenser and wall temperature |
| $Q$, $Q_a$, $Q_s$ | Flows, aerosol and sample flow |
| $z$, $r$, $z'$, $r'$ | Axial z and radial r direction inside a tube. Normalized axial $z' = z/R_t$ and radial $r' = r/R_t$ direction |
| $R_t$ | Radius of the tube (either condenser, insulation or saturator tube radius) |
| $p_v$, $p_{sat}$ | Partial and saturation vapor pressure |
| $S$ | Saturation ratio $S = p_v/p_{sat}$ |
| $Pe$, $Re$, $Pr$, $Sc$ | Péclet, Reynolds, Prandtl and Schmidt number (dimensionless) |
| $D_{K,eq}$ | Equilibrium Kelvin diameter |