# Peer review of "Pressure-dependent performance of two CEN-specified Condensation Particle Counters"

_EGUsphere, 2022_

## Author Response (AR1)

**RC1: 'Comment on egusphere-2022-1206', Anonymous Referee #1, 23 Nov 2022**

The manuscript by Bauer et al. presents experimental and simulation characterization of two CPC models that fulfil the CEN technical standard. To my understanding, characterization of the CEN CPCs is the main progress of this paper, other methods and qualitative understandings on the instruments are previously known.

The size and dependent counting efficiency curves were measured and simulated at pressures ranging from 1 to 0.15 bar, and the cutoff diameter was found to increase with decreasing pressure, while the plateau counting efficiency was found to decrease with decreasing pressure. The pressure dependent effects were explained by the simulations, demonstrating that the length of the insulator between the saturator and condenser affects the results. The manuscript is well-written and the experiments and simulations appear valid, and I have only very minor technical suggestions for the manuscript.

We want to thank the referee for the valuable comments on the manuscript which help to improve the manuscript. We will address the comments as described in our following responses in blue.

- intro: I would take the opportunity in this manuscript to briefly summarize some of the past findings related to the pressure dependency of a CPC. Currently, only some recent studies are vaguely mentioned. It would give better introduction to the reader on this topic

  We changed the introduction in accordance with the remarks from RC1 and RC2. We included some sentences on past findings related to pressure dependencies of CPCs. We referenced following papers on this topic:

  Cofer, W. R., Anderson, B. E., Winstead, E. L., and Bagwell, D. R.: Calibration and demonstration of a condensation nuclei counting system for airborne measurements of aircraft exhausted particles, 32, 169–177, https://doi.org/10.1016/S1352-2310(97)00318-X, 1998.

  Dreiling, V. and Jaenicke, R.: Aircraft measurement with condensation nuclei counter and optical particle counter, 19, 1045–1050, https://doi.org/10.1016/0021-8502(88)90097-3, 1988.

  Heintzenberg, J. and Ogren, J. A.: On the operation of the TSI-3020 condensation nuclei counter at altitudes up to 10 km, 19, 1385–1387, https://doi.org/10.1016/0004-6981(85)90268-9, 1985.

  Noone, K. J. and Hansson, H.-C.: Calibration of the TSI 3760 Condensation Nucleus Counter for Nonstandard Operating Conditions, 13, 478–485, https://doi.org/10.1080/02786829008959462, 1990.

  Saros, M. T., Weber, R. J., Marti, J. J., and McMurry, P. H.: Ultrafine Aerosol Measurement Using a Condensation Nucleus Counter with Pulse Height Analysis, 25, 200–213, https://doi.org/10.1080/02786829608965391, 1996.

  Schröder, F. and Ström, J.: Aircraft measurements of sub micrometer aerosol particles ( > 7 nm) in the midlatitude free troposphere and tropopause region, 44, 333–356, https://doi.org/10.1016/S0169-8095(96)00034-8, 1997.

  Seifert, M., Tiede, R., Schnaiter, M., Linke, C., Möhler, O., Schurath, U., and Ström, J.: Operation and performance of a differential mobility particle sizer and a TSI 3010

condensation particle counter at stratospheric temperatures and pressures, 35, 981–993, https://doi.org/10.1016/j.jaerosci.2004.03.002, 2004.

Weigel, R., Hermann, M., Curtius, J., Voigt, C., Walter, S., Böttger, T., Lepukhov, B., Belyaev, G., and Borrmann, S.: Experimental characterization of the COndensation PArticle counting System for high altitude aircraft-borne application, 2, 243–258, https://doi.org/10.5194/amt-2-243-2009, 2009.

- Line 82, η∞, i would use something else than infinity. Already at ~5 µm the detection efficiency starts to drop.

  We want to thank the referee for pointing this out. We changed our nomenclature from η∞ to η_plat throughout the entire manuscript and we changed the definition to η_plat = η(d_p,50 << d_p  < 1µm) to account for the effect of the decreasing detection efficiency at sizes > 1 µm.

- L285, please include the inputs for loss calculations, I cannot preproduce your loss calculation results as they are now presented.

  We added a section in the Supp. Info with the parameters for the loss calculation.

- L391-393, the temperature difference is 1 degree. I would say they are within the experimental/fitting uncertainties

  We removed this sentence after recomputing some of the graphs. See additional comment at the end.

**RC2: 'Comment on egusphere-2022-1206', Alfred Wiedensohler, 11 Feb 2023**

Dear authors,

We want to thank the referee for the valuable comments on the manuscript which help to improve the manuscript. We will address the comments as described in our following responses in blue.

I would like to suggest a change in the scope of the article.

The CEN/TS for the condensation particle counters is currently under revision and will become a new EU standard by 2024. In the new EU standard, the section for the obligatory calibration at two different pressures will be cancelled. The DP50 detetction diameter will be shifted to 10 nm.

We are aware that the CEN/TS16976 is currently under revision and will become a European standard soon. However, the CPCs used in this study were manufactured to meet the CEN Technical Specification 16976:2016, i.e. they for example have a DP50 detection diameter of 7 nm. In addition, the study was undertaken while the CEN Technical Specification 16976:2016 was and is still valid. This is also reflected in our nomenclature of calling the CPC "CEN-specified" and not "CEN-standardized". Since some parameters might be changed in the standard, we included Sec. 1.2 (CEN Technical Specification 16976:2016), where we describe the main points of the technical specification so that it is clear to the reader what we are referring to. Even if the parameters will change in the future, it will not alter our measurement results.

To point out, that CEN/TS16976 will become a European standard, and that parameters of the technical specification will change in the future, we added the following sentences to the Sec. 1.2:

"[…] The CEN/TS16976 will become a European standard in the near future and some parameters (e.g. the cut-off diameter dp,50 or pressure calibration) might change. However, when the publication was written the CEN/TS16976:2016 was still valid and the most relevant specifications for the presented study are summarized in the following paragraphs: […]"

The TSI 3772 is not sold anmore by the company TSI.

With this paper, we wanted to show a deeper physical understanding of low-pressure behavior of modern CPC and selected two examples of CEN-CPCs, the TSI 3772-CEN and the Grimm 5410 CEN. We are aware that the TSI 3772 and TSI 3772-CEN CPCs are not sold any more. However, many TSI 3772 and TSI 3772-CEN CPCs are still used in many labs and in particular on several research aircraft. We therefore think that our results on the pressure-dependent performance of these CPCs are still interesting to the scientific community. In addition, to our knowledge, the TSI 3772 and TSI 3772-CEN have significant internal differences (e.g. pulse height monitoring). Therefore, it is important to mention that the tested CPCs are the CEN version.

In meanwhile, there was also a finding that there is a gap of few percent in counting efficiency at the plateau (40 nm) between counting pulses and measuring the electric current with the electrometer. The CEN working group found an agreement now. The solution is to define a correction factor for this gap, since the ISO working group decided that the electrometer measurement is the true value. Each counter (or model) will use a gap correction factor, either for an individual counter for a counter model. It is from the manuscript however not clear how you got approximately 100% detection efficiency at the plateau around 40 nm.

We have discussed this point with the manufactures of the instruments. As we understand, the gap is only relevant if the pulse output is used for analysis. However, for this current study we have not used the pulse output. In Sec. 2.2. in L193 we wrote: "Both CPC-models come with an internal

coincidence correction (e.g. live-time correction for TSI 3772 CEN), which is why we chose the corrected concentration output of the CPCs (NCPC) for our data analysis."

I propose following, a) remove the focus on a CEN-compliant particle counter, [...] c) change the focus and title to the behaviour at low pressures of modern particle counter,

This study is not focusing on CEN specifications but rather on the behavior of CEN-specified CPCs under varying pressure conditions. The starting point of our investigations were the two different CPC models which are CEN-specified under the currently valid CEN/TS16976:2016. When we bought these CPCs they came with a CEN certificate from the manufacturer. Despite their design differences, they have almost the same performance (cut-off curves) under standard conditions as reported in their calibration certificate. However, the performance (cut-off curves) changes under low-pressure conditions which is one of the points we wanted to illustrate with this publication. We will change the title to: "Pressure-dependent performance of two CEN-specified Condensation Particle Counters" to point out that we used two different types of CEN-specified CPCs.

We also changed the introduction to take the RC1 and RC2 comments into account.

b) define your actually measurements of the plateau efficiency as 100% as a base for the modeling of the counting efficiency at lower pressures

No pulse output data were used for the analysis of our measurements. We would like to refer to our reply to the "gap question" above. In the manuscript, the calculated activation efficiency and why it reaches 100% is explained in L382 and following.

d) relate this modeled results to experimental results of previous publications of more elderly particle counters. The design of the saturator and condenser might have been changen during the last 30 years.

One key feature of our simulations is the modeling of the insulator before the condenser, which is important for understanding the low-pressure behavior. The only other publication we are aware of that includes the insulator is following:

Reinisch, T., Radl, S., Bergmann, A., Schriefl, M., and Kraft, M.: Effect of model details on the predicted saturation profiles in condensation particle counters, 30, 1625–1633, https://doi.org/10.1016/j.apt.2019.05.011, 2019.

For confirmation, we compared our simulation results to their results. However, the authors have used a CPC with 23nm cut-off diameter, which is different from the settings in our study.

Because our simulations include the insulator, it is questionable to draw comparisons to previous simulation results which do not include the insulator.

In addition, the dimensions for other CPC types (especially the dimensions of the insulator between the saturator and the condenser) are often a confidential information and are not mentioned in publications. Therefore, it is difficult to compare the results of our model simulations with experimental result of previous publications of more elderly particle counters.

best regard Alfred Wiedensohler

We have to make a comment and a small correction to our initial draft of this publication.

While revising the simulation code for another project, we realized that the temperature field was not computed correctly which resulted in some small deviations to the resulting 2D fields. We corrected it and replotted all the figures that are affected (Fig. 4, Fig. 6, Fig. 7 and Table 3). Fig. 5 is not affected because the temperature was calculated correctly for this plot.

As a consequence, the simulated activation and counting efficiency curves were shifted slightly towards smaller diameters and the simulated curves and onset diameters agree now very well with the measurements. Therefore, we have also removed following sentences form the manuscript:

Line 375 to 381:
Interestingly, the onset diameters $d_{p,0}$ of both CPC-models are in general lower for the measured counting efficiency curves (see Table 2) compared to the numerically calculated ones (Table 3). The difference between the calculated and measured onset diameter might be explained by the chemical interplay of butanol nucleating onto silver particles, which is responsible for the activation of particles smaller than the Kelvin-diameter (Eq. (9)) (Tauber et al., 2019). This chemical effect is depending on the nucleation temperature and is increasing with decreasing nucleation temperature. Further investigations and a parametrization are needed to incorporate this chemical effect into simulations.

And Line 391 to 394:
For the Grimm CPC, there is a larger deviation, which might be explained by the chemical effect on the onset diameter, because in the Grimm CPCs the nucleation temperature is slightly lower and therefore the effect is more relevant (Tauber et al., 2019). For both CPCs, at 500 hPa the parameters and counting efficiency curves of the simulations fit best with those of the measurements.

Removing the last line is also in accordance with the comment of Referee #1 stating: "L391-393, the temperature difference is 1 degree. I would say they are within the experimental/fitting uncertainties"

---

## Author Response (AR2)

Dear Editor,

following your comment, we double-checked all values of Table 2. We couldn't find an error in the table.

For the Grimm CPC at 150 hPa the cut-off diameter dp_50 is 21.5 nm. The dp_50 is the diameter where the counting efficiency curve reaches the value of 50%. Because the plateau efficiency is only 54.6% for this pressure stage the dp_50 is shifted quite a bit to larger diameters compared to dp_50_fit. A horizontal 0.5 line would intersect at 21.5nm with the 150 hPa Grimm counting efficiency curve at in Fig. 3.

Thank you for pointing this out and taking a very detailed look at our manuscript!
We also want to thank the entire editorial team for helping us with the submission.

Best wishes,
Bernadett Weinzierl and Paulus Bauer on behalf of the authors